# THE DATA-QUALITY ILLUSION: RETHINKING CLASSIFIER-BASED QUALITY FILTERING FOR LLM PRETRAINING

## ABSTRACT

Large-scale models are pretrained on massive web-crawled datasets containing documents of mixed quality, making data filtering essential. A popular method is Classifier-based Quality Filtering (CQF), which trains a binary classifier to distinguish between pretraining data and a small, high-quality set. It assigns each pretraining document a quality score defined as the classifier's score and retains only the top-scoring ones. We provide an in-depth analysis of CQF. We show that while CQF improves downstream task performance, it does not necessarily enhance language modeling on the high-quality dataset. We explain this paradox by the fact that CQF implicitly filters the high-quality dataset as well. We further compare the behavior of models trained with CQF to those trained on synthetic data of increasing quality, obtained via random token permutations, and find starkly different trends. Our results challenge the view that CQF captures a meaningful notion of data quality.

## 1 INTRODUCTION

Large-scale models are pretrained on large amounts of data, and the quality of these data is a critical factor in achieving state-of-the-art performance. Among various heuristics for leveraging data quality to improve on downstream tasks, Classifier-based Quality Filtering (CQF) is recognized as a cornerstone of data processing. CQF has now become widely adopted and is, for instance, part of established pretraining pipelines like those of GPT3 (Brown et al., 2020), LLama (Touvron et al., 2023), and PALM (Chowdhery et al., 2023). It is also a key component of several widely used public datasets, such as DCLM (Li et al., 2024) or the SmolLM corpus (Ben Allal et al., 2024).

CQF, as illustrated in Figure 1, trains a binary classifier to distinguish documents from a large, low-quality pretraining set (LQ set) from those of a small, high-quality dataset (HQ set). It then assigns a scalar quality score to each document within the LQ set, defined by the classifier's score. The filtered dataset is formed by selecting the top $k$ fraction of documents in the pretraining set, ranked by their quality score.

The goal of this paper is to understand the mechanics behind CQF, its impact on downstream performance, and to challenge the underlying notion of quality it defines. Concretely, *does CQF truly select data that resemble the HQ set, as it is commonly believed? Does the quality score it incurs match the general intuition about data quality?*

We start by highlighting a paradox in how CQF works: although CQF consistently improves performance on downstream tasks, it does not necessarily improve language modeling on the HQ set. This finding challenges the widely held belief that CQF improves models by selecting training data that are similar to the HQ data. We explain this paradox by the fact that CQF is akin to an implicit quality filtering of the *HQ set* itself, which upweights data in the HQ set that are far from the LQ set. This means that models trained with CQF are not necessarily good at language modeling on the whole HQ set, but rather on a higher-quality subset of it. Moreover, we show that this filtering of the HQ set aligns with downstream tasks for most choices of HQ sets, which explains the paradox. We then compare CQF to importance sampling methods (Xie et al., 2023; Grangier et al., 2024), which explicitly attempt to resample the LQ set to follow a distribution close to the HQ set. We highlight a

Figure 1: **Classifier-based Quality Filtering (CQF) pipeline**. A document embedding model (e.g. sBert, Artic-Embed or FastText) embeds documents from a high-quality dataset and the pretraining set. A binary classifier is trained on those embeddings to distinguish the HQ set from the pretraining set. Scores assigned by the classifier are used to rank documents from the pretraining set. The top $k$ fraction of those documents constitutes the new filtered CQF dataset.

stark difference between the two methods: importance sampling yields better language modeling on the HQ set, but it does not benefit from the aforementioned implicit filtering of the HQ set.

Beyond these paradoxes, we introduce a new lens to probe whether CQF induces a meaningful notion of quality. Specifically, we formalize the notion of *data conditioning*: along a true quality axis, training on "clean" data should give better performance on "dirty" test distributions than training directly on the dirty distribution. This behavior fundamentally depends on the optimization algorithm used to approximately minimize the training loss. Indeed, if the training algorithm were perfect, training on the dirty data itself would always yield the best possible loss on that very data. Therefore, this phenomenon can only arise because optimization on the clean dataset is easier, hence the term data conditioning. We demonstrate that this desirable property is clearly observed when constructing datasets with ground-truth mixtures of clean and dirty documents, as inspired by Kallini et al. (2024). In contrast, subsets selected by CQF fail to exhibit any such data-conditioning ordering, suggesting that the notion of quality CQF captures is more limited and closely related to stylistic or domain similarity—contexts in which "training cleaner" does not universally help.

## 1.1 RELATED WORK

Recent surveys (Albalak et al., 2024; Longpre et al., 2024) provide comprehensive overviews of data selection pipelines and identify classifier- and perplexity-based filtering as the most widely used techniques, with classifier-based methods being the most effective in practice (Li et al., 2024). A common underlying assumption across these approaches is that pretraining on data resembling a small, trusted high-quality (HQ) set (e.g., Wikipedia, books, curated instructions) improves downstream performance. This belief has motivated two main strategies that operate at the document level: directly mimicking the HQ distribution via importance sampling or indirectly approximating it through classifier-based filtering. In the importance sampling paradigm, Xie et al. (2023) approximate the likelihood ratio between HQ and LQ data to guide resampling of the LQ set, while CRISP (Grangier et al., 2024) uses clustering of the pretraining data to best match the HQ set.

CQF, on the other hand, uses a classifier to score LQ documents by learning boundaries between HQ and LQ samples. CQF is widely adopted in state-of-the-art pipelines: GPT-3 (Brown et al., 2020) employs a classifier with Pareto-biased sampling; LLaMA (Touvron et al., 2023) filters Common Crawl using Wikipedia as HQ; GLaM (Du et al., 2022), PaLM (Chowdhery et al., 2023), and RedPajama (Weber et al., 2024) similarly rely on Wikipedia and books. More recently, Li et al. (2024) introduced DCLM, a large-scale filtered dataset centered on CQF, using ELI5 (Fan et al., 2019) and OpenHermes (Lian et al., 2023) as HQ sources. Wang et al. (2025) study methods to build HQ sets, and Soldaini et al. (2024) propose the Dolma Toolkit, featuring CQF that is applied to the Dolma dataset iteself. RefinedWeb (Penedo et al., 2023) and FineWeb (Penedo et al., 2024) use classifiers to extract English documents. Artic-Embed (Merrick et al., 2024) is a popular document embedder for training quality classifiers, underlying Python-edu and FineWebEdu (Ben Allal et al., 2024) datasets. Recently, Mizrahi et al. (2025) analyzed how aggressive filtering should be as function of model and data scales. Finally, classifiers can also be used to filter toxic content (Welbl et al., 2021).

| Dataset | Number of Documents | Source |
|---|---|---|
| OpenOrca (Lian et al., 2023) | 3M | GPT-4/GPT-3.5 |
| ELI5 (Fan et al., 2019) | 325k | Reddit |
| OpenHermes (Teknium, 2023) | 240k | GPT-4 |
| KnowledgePile (Fei et al., 2024) | 1M | Scientific blog & papers |
| openwebmath (Paster et al., 2023) | 6.3M | Mathematical webtext |
| ARC Easy (Clark et al., 2018) | 2.25k | grade-school level MCQA |

Table 1: **Overview of the "high-quality" datasets used for CQF in our study.**

Beyond CQF and importance sampling, recent works learn proxy scores directly linked to downstream performance rather than assuming and imposing any fixed notion of quality. For example, Mizrahi et al. (2025) train regressors to predict closeness to evaluation tasks, Zhuang et al. (2025) combine multiple quality dimensions into learned mixtures, and older methods rely on LLM perplexity (Wenzek et al., 2020). These methods suggest that the best data may not necessarily resemble a specific HQ corpus, but rather satisfy task-relevant criteria that can be discovered during training.

## 2 CLASSIFIER-BASED QUALITY-FILTERING

We describe the Classifier-based Quality-Filtering (CQF) method as it is used in the literature and in this paper. CQF takes as inputs a high-quality (HQ) dataset, $D_{HQ}$, a pretraining dataset that is generally of low quality, $D_{LQ}$, and a selection fraction $k$ between 0 and 100%.

**Low-quality (LQ) dataset.** This is a standard pretraining set, which, in the context of LLM pre-training, contains curated documents gathered from a large web crawl spanning diverse data sources. While the dataset is huge—containing enough tokens to train large models without repetitions—it also includes many low-quality, badly formatted, or uninformative documents. The overall goal of data selection is to select a subset of this LQ set that leads to better model performance. In this paper, we take RedPajama-V2 as our LQ set, which contains 32T tokens.

**High-quality (HQ) dataset.** This is a high-quality dataset made of documents from a highly curated source. These documents are well formatted, have relevant content and are sometimes manually annotated. They can be data coming from proofread websites such as Wikipedia, or sentences generated by a sufficiently good language model. However, the HQ dataset is typically quite small and insufficient on its own to train a model. Instead, it serves two key purposes to guide the data selection process: 1) as a target for selection, where data in the LQ set that resemble the HQ set are considered high quality, and 2) as a benchmark to evaluate the effectiveness of data selection, with models achieving low loss on this dataset considered to be performing well. Table 1 gives an overview of HQ sets used in this work.

CQF is a widely used method for data selection that filters data from the LQ set, guided by the HQ set. We now describe its practical implementation, which is illustrated in Figure 1.

**Embedding.** Each document in the HQ and LQ datasets is embedded in a vector space $\mathbb{R}^p$. Since the whole LQ set has to be embedded, the embedding method needs to be scalable. In practice, we use sBert, with $p = 384$. Another popular choice is FastText (Joulin et al., 2016).

**Classifier training.** A training set made of $n$ embeddings from the HQ set and $n$ others from the LQ set is used to train an L2-regularized logistic regression. The regularization coefficient is taken as the one maximizing accuracy on a held-out set. Once this classifier is trained, it defines the **CQF score** function $s(x) \in [0, 1]$, that, for any document $x$, defines a scalar that measures how likely the classifier is to identify this document as a member of the HQ set. This score $s(x)$ is often called quality signal (Weber et al., 2024), which is why, in the context of CQF, we will refer to it as **quality of document** $x$. A goal of this paper is to understand whether this definition of quality is appropriate.

**"Quality" filtering.** In order to estimate the distribution of the scores on the LQ dataset, a subset of the LQ dataset is scored, which allows us to estimate the cumulative density $C(\tilde{s}) = \mathbb{P}(s(x) \leq \tilde{s} | x \in D_{LQ})$ for all $\tilde{s} \in [0, 1]$. Then, for a given **selection fraction** $k$, only the top $k$ fraction of documents in the LQ set is kept, resulting in a filtered dataset $D_{CQF} = \{x \in D_{LQ} | C(s(x)) \geq 1 - k\}$.

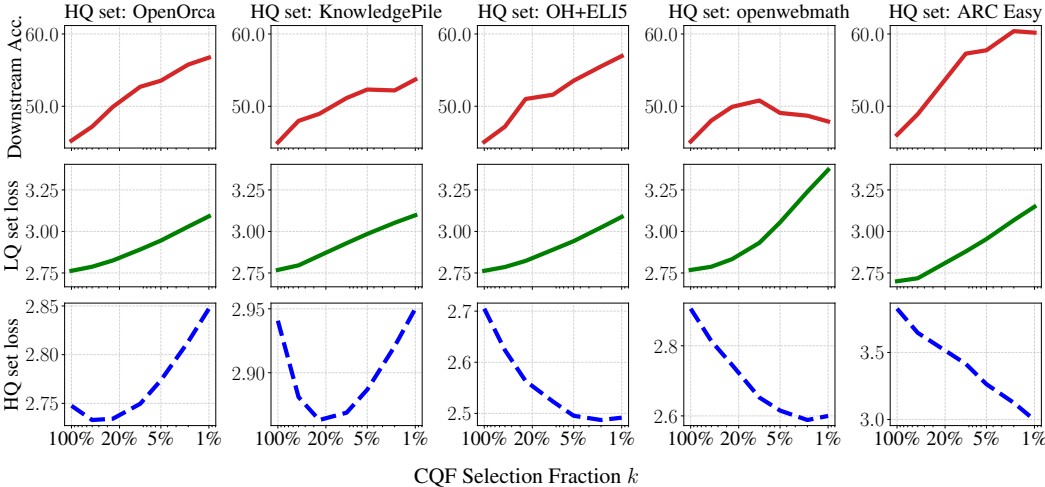

Figure 2: **Top row**: Models trained on increasingly selective data show improved performance on downstream tasks. **Bottom row**: When evaluated on the HQ dataset itself, these models do not necessarily improve as there is a non-increasing relationship between downstream performance and loss on the HQ set. The performance of a model trained on the LQ set is given by the leftmost point in each figure, corresponding to $k = 100\%$

This selects the documents in the LQ set that are most likely to belong to the HQ set, based on the score defined by the classifier, and are therefore "higher-quality" documents. This dataset is then used to train models in place of the low-quality dataset. One clear limitation of CQF is that the number of training tokens available in the dataset is $k \times D$ where $D$ is the total number of tokens in the LQ set. Too small values of $k$ lead to scarce datasets on which models cannot be trained without repeating data or even overfitting. In this paper, we step away from this limitation and always use values of $k$ such that there are enough data in $D_{\text{CQF}}$ to train a model without repeating data. This allows us to focus solely on the impact of data quality rather than on the effects of repeated training examples.

Table 2: Hyperparameters used for training models

| Size | 125M | 350M | 1.3B |
|---|---|---|---|
| *Architecture* | | | |
| Vocab Size | 32K | 32K | 32k |
| Embedding dim. | 768 | 1,024 | 2,048 |
| Latent dim. | 3072 | 4,096 | 8,192 |
| Num. heads | 16 | 16 | 16 |
| Depth | 12 | 24 | 24 |
| Context lenght | 1,024 | 1,024 | 1,024 |
| *Optimization* | | | |
| Batch size (tokens) | 115K | 32K | 115K |
| Learning rate scheduler | lin. decay | lin. decay | lin. decay |
| Learning rate peak | $1e^{-4}$ | $1e^{-4}$ | $1e^{-4}$ |
| Grad clipping | 5.0 | 5.0 | 5.0 |
| Steps | 64K | 256K | 1M |
| Num. train tokens | 8B | 8B | 120B |

**Evaluations.** After pretraining, models are evaluated by scoring them on evaluation benchmarks, such as general knowledge question answering. Performance on these datasets is indicative of the usefulness of models after post-training. In this work, we consider evaluations on ARC-Easy, ARC-Challenge, MMLU, and reward-bench. The bulk of our experiments is done on ARC-Easy, which has better-than-random performance at small scales. Model architectures are found in Table 2.

## 3 CQF IMPROVES MODEL EVALUATIONS

We begin with the observation that motivates the wide adoption of CQF. We train 350M models on CQF datasets with different HQ datasets and values of $k$. We then evaluate those models by computing their accuracy on ARC-Easy. We also use ARC-Easy itself as the HQ set. We display the results in Figure 2, top row. Among all HQ sets, using ARC-Easy leads to the best downstream performance. We observe that the performance on the downstream task generally improves as we select datasets of higher quality, with smaller values of $k$. This occurs for OpenOrca, KnowledgePile, OH+ELI5, and ARC-Easy, but for openwebmath, we observe a performance dip if we select a value of $k$ that is too small. A simple explanation is that CQF with openwebmath selects too specialized

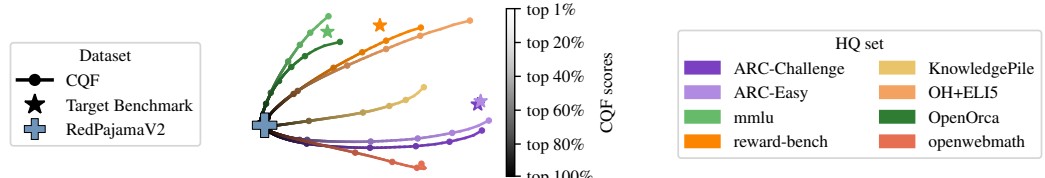

Figure 3: **Two-dimensional PCA projections of sBert embeddings from quality buckets defined by classifiers, each using a different HQ set..** Quality buckets across classifiers (CQF) used in the literature exhibit alignment towards benchmark datasets. When considering the top 100%, we fall back to the original pretraining dataset (RedPajama-V2) regardless of the HQ set used.

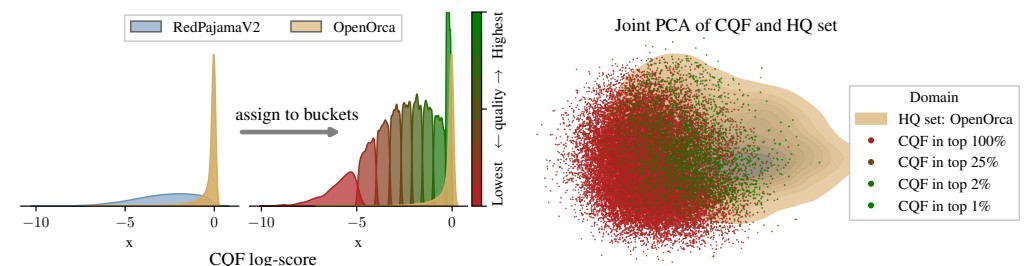

Figure 4: **CQF works by filtering out the low-quality data (red),** *not* because the retained data (green) resemble the HQ set (orange). This is clear both from the raw log-scores of the classifier (**left**), and in 2D PCA of the sBert latent space (**right**). TSNE show similar patterns in Appendix C.

documents. We confirm this alignment between data selected by CQF and common benchmarks in Figure 3 by examining a 2D PCA of their latent space.

## 4 CQF DOES NOT SELECT DATA THAT RESEMBLE THE HIGH-QUALITY SET

**CQF ranks data based on likelihood ratios.** Assuming that the binary classifier trained in CQF is Bayes-optimal, the CQF quality score of a document $x$ is $s(x) = \frac{p_{\text{HQ}}(x)}{p_{\text{HQ}}(x)+p_{\text{LQ}}(x)}$ (Hastie et al., 2009). As such, scores are an increasing function of the *density ratio*: $s(x) = \phi\left(\frac{p_{\text{HQ}}(x)}{p_{\text{LQ}}(x)}\right)$ with $\phi(t) = \frac{t}{t+1}$. The ordering of documents implicitly defined by CQF is therefore that of the likelihood ratio: a document $x$ is of "higher quality" than a document $y$ if $\frac{p_{\text{HQ}}(x)}{p_{\text{LQ}}(x)} \geq \frac{p_{\text{HQ}}(y)}{p_{\text{LQ}}(y)}$. This contrasts with the "importance sampling" ranking, which would rank $x$ higher than $y$ solely based on their likelihood under the HQ distribution, i. e., if $p_{\text{HQ}}(x) \geq p_{\text{HQ}}(y)$. A simple conclusion is that, since in general the LQ set is not uniformly distributed, CQF does not select samples that are most likely to come from the HQ set only. Instead, it prefers documents that are both likely under the HQ distribution (high $p_{\text{HQ}}(x)$) and unlikely under the LQ distribution (low $p_{\text{LQ}}(x)$). In other words, with CQF, data are filtered based on a trade-off between being close to the HQ set and far from the LQ set. This phenomenon is clear when plotting the score densities of data filtered by CQF in Figure 4.

### 4.1 KULLBACK-LEIBLER DIVERGENCE BETWEEN DATASETS

For each model trained in section 3, we also compute its next-token prediction loss on the HQ set (Figure 2, bottom row). We observe U-shaped curves for all HQ datasets except ARC-Easy. For these HQ sets, the optimal $k$ that yields the smallest loss is often large. Remarkably, small values of $k$ can result in models that perform even worse on the HQ set than a model trained on the full LQ set, as seen with OpenOrca or KnowledgePile. This behavior contrasts with using ARC-Easy as HQ set, where reducing $k$ consistently improves both model performance and language modeling.

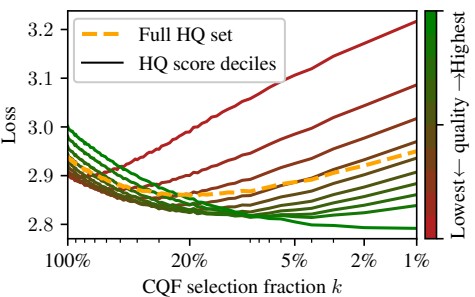 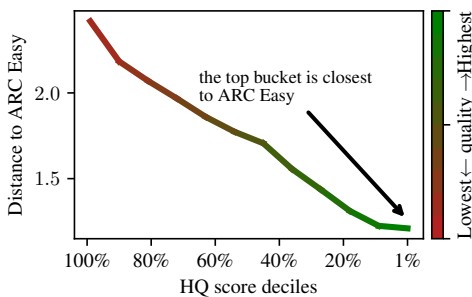

Figure 5: **CQF implicitly filters the HQ set**. We split the HQ set (KnowledgePile) into 10 deciles of CQF scores. **Left**. For each model trained with CQF at a given fraction $k$, we report the loss of the model on each of these 10 deciles. The reddest curve corresponds to the loss on the HQ elements with the bottom $10\%$ scores, while the greenest curve corresponds to the top $10\%$. Our findings indicate that only the high-quality deciles of the HQ set exhibit a decreasing loss. This suggests that the classifier effectively identifies and learns the features within these deciles, enabling the models to make better predictions. However, on average over all the deciles (dotted line), the loss is a U-curve, recovering the loss in Figure 2 (second row and column). **Right**. In sBert latent space, we compute the distance between the barycenter of ARC-Easy to the barycenter of each HQ decile. This distance correlates well with performance on the ARC-Easy benchmark itself.

As a result, there is a clear discrepancy between the loss on the HQ set—which reflects how closely the pretraining data resemble the HQ distribution—from the achieved downstream performance (ARC-Easy). This challenges the standard belief that CQF filters data to get closer to the HQ set.

**Loss on the HQ set as a proxy for the distance between CQF and HQ set.** The loss measured on the HQ set can be interpreted as a measure of how *different* the filtered data are from the HQ set in terms of Kullback-Leibler (KL) divergence, under the assumption that the model has infinite capacity (Cover, 1999). Indeed, in this case, the model's parameters $\theta$ are such that the model trained on the filtered set by CQF would perfectly represent its data distribution, i.e., $p_\theta(x) \approx p_{\mathrm{CQF}}(x)$.

Evaluating this model on the HQ set yields a next-token prediction loss equal to $\mathbb{E}_{x \sim D_{\mathrm{HQ}}}[-\log p_{\mathrm{CQF}}(x)]$. This quantity can be decomposed as,

$$\mathrm{H}(D_{\mathrm{HQ}}) + \mathrm{KL}(D_{\mathrm{HQ}} \| D_{\mathrm{CQF}}),$$

where $\mathrm{H}(D_{\mathrm{HQ}})$ is the entropy of the HQ distribution (a constant when changing $k$), and $\mathrm{KL}(D_{\mathrm{HQ}} \| D_{\mathrm{CQF}})$ is the KL divergence from the HQ distribution and the distribution of data filtered by CQF. Hence, under the hypothesis that the models trained in these experiments accurately represent $p_{\mathrm{CQF}}$, the observed increase in HQ loss for small $k$ means that the corresponding pretraining sets diverge further away from the HQ distribution. To our knowledge, this phenomenon has not been previously identified. In the next section, we investigate the reasons behind it.

### 4.2 CQF IMPLICITLY FILTERS THE HIGH-QUALITY DATASET AS WELL

One way to interpret the CQF selection rule is that it is a reweighting of the distribution of the HQ set, with non-uniform weights: it puts a larger weight on documents that are far from the LQ set.

As a result, CQF can be understood as 1) selecting data in the HQ set that are far from the LQ set and then 2) selecting data in the LQ set that are close to *that* portion of the HQ set. To validate this interpretation, we further partition the HQ set itself into 10 "quality" buckets according to their CQF scores. We then measure the next-token prediction loss on these 10 domains achieved by models trained with CQF by varying $k$ in Figure 5. Interestingly, the loss on the top-scoring documents from the HQ set behaves very differently than the loss on the bottom-scoring data from the same set. More precisely, the loss on the top-scoring HQ data is monotonic with $k$, while the loss on the bottom-scoring HQ data rises sharply as $k$ decreases. This analysis decomposes the overall U-shaped loss reported in the previous section into the average loss across different quality levels within the HQ set. Notably, this implicit filtering of the HQ set itself is beneficial. In fact, data in the HQ set that

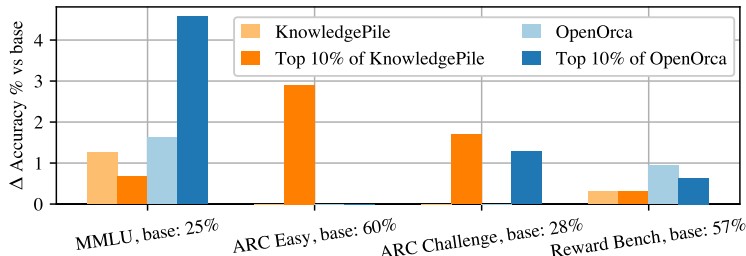

Figure 6: **Finetuning a 1.3B model on HQ sets and on their top decile**. We report the best performances during fine-tuning; no bar means that fine-tuning on that set does not improve the performance on that benchmark. Darker colors indicate finetuning on the top 10% of the HQ set according to the CQF classifier, while light colors indicate finetuning on the whole HQ set. The same number of tokens is used in both scenarios. OpenOrca aligns most closely with MMLU, whereas KnowledgePile shows stronger alignment with ARC-Easy, supporting the trend observed in Figure 3.

resemble data from the LQ set are likely to be of lower quality, since the LQ set contains a significant amount of noisy data. This resolves the earlier paradox: top-scoring data within the HQ set are more aligned with the evaluation task than those from the HQ set with lowest scores; see Figure 5.

We further validate that these top deciles of HQ sets are aligned with downstream evaluations by finetuning a 1.3B model on them, as well as on the full HQ set. We report the corresponding gains in accuracy in Figure 6. This again shows that the top decile of KnowledgePile is aligned with ARC-Easy, while the full set is not. We now formalize this implicit filtering intuition.

**CQF as a reweighting of the HQ set.** Letting $r(x) = \frac{p_{\mathrm{HQ}}(x)}{p_{\mathrm{LQ}}(x)}$ be the likelihood ratio, CQF selects data in the LQ set such that $r(x) \geq \tau$, where $\tau$ is calibrated so that only a fraction $k$ of the LQ set is selected. The CQF dataset's density can be rewritten as

$$p_{\mathrm{CQF}}(x) = \frac{1}{Z} 1_{r(x) \geq \tau} p_{\mathrm{LQ}}(x) = w(x) p_{\mathrm{HQ}}(x), \text{ where } w(x) \propto \frac{1_{r(x) \geq \tau}}{r(x)}, \tag{1}$$

which means that it is a reweighted version of the HQ set density, with weights $w(x)$, and where $Z$ is a normalization constant. The most upsampled points in the HQ set, which have a high value $w(x)$, are therefore those such that $r(x)$ is above $\tau$ while being small. This is akin to a filtering of the HQ set based on the likelihood ratio value $r(x)$. This explains the results in Figure 5: as the fraction $k$ reduces, $p_{\mathrm{CQF}}$ gets close to a filtered version of $p_{\mathrm{HQ}}$ where only top-scoring samples are kept.

## 5 CQF IS NOT IMPORTANCE SAMPLING

A common belief behind the use of CQF is: "Ideally, we would train on the HQ set, but we don't have enough data. So we use CQF to mimic data from the HQ set." As we have seen in the previous section, assuming that the classifier is Bayes-optimal, CQF draws samples from the LQ set following the density $w(x) p_{\mathrm{HQ}}(x)$, where $w(x)$ is not uniformly equal to 1. On the other hand, importance sampling methods try to sample elements from the LQ set that directly follow the density $p_{\mathrm{HQ}}$. We use the CRISP method (Grangier et al., 2024) in order to implement importance sampling, with the same models as in section 3, with OpenOrca and ARC-Easy as HQ sets. We report the loss on the HQ set and the downstream accuracy in Figure 7, as well as those of the models trained with CQF. OpenOrca being diverse and multi-topic, we found that $C = 4096$ clusters are sufficient to capture that distribution

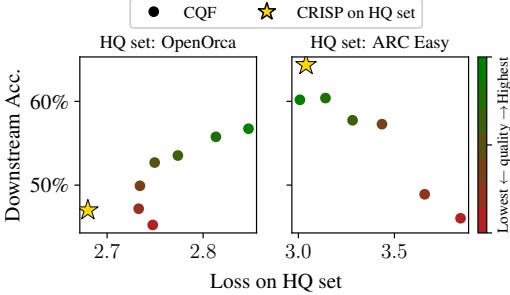

Figure 7: **Performance comparison between CQF and importance sampling-based approach (CRISP)**. CQF induces a data selection that is substantially *different* from the HQ set. Colors indicate more (green) or less (red) filtering.

well, whereas ARC-Easy requires $C = 260k$ clusters. We observe that importance sampling indeed leads to good language modeling on the HQ set, which translates to better downstream performance when the HQ set is the downstream task itself (right), but not when the HQ set is a curated dataset (left). In that case, CQF leads to better downstream performance than importance sampling.

**Training with CQF can be better than training on an infinite HQ set** A widely spread idea about the use of CQF is that if we had enough data from the HQ set, we would train models on the HQ set itself. We now argue that this is not necessarily the case and that training on the CQF set can be better than on the HQ set, even with limited data from the HQ set. To do so, we use a base HQ set with limited tokens, $D_{\text{HQ}}^{\text{base}}$; in practice, we use OpenOrca. We define a new HQ set as $D_{\text{HQ}} = \text{CQF}(D_{\text{HQ}}^{\text{base}}, 20\%)$, i.e. the CQF with $k = 20\%$ of the base HQ set. Now, this new high-quality set contains enough tokens to train large-scale models, since it contains 20% of the LQ set. We then construct a new CQF set using that new HQ set, and train a model on the top 1% of data. We observe that the accuracy of a model trained on the HQ set is **50.1**%, while the accuracy of the new model trained on the CQF set is **53.8**%. In summary, we obtain better models by using CQF than by training directly on the HQ set, illustrating once again the benefits of the implicit filtering provided by CQF. This opens a promising avenue for data selection, where CQF is used as a way to improve on the HQ set.

## 6 DISCUSSION: DOES CQF DEFINE A SOUND NOTION OF QUALITY?

The goal of this section is to offer a different perspective on the concept of quality by introducing a formal definition based upon optimization considerations. Within this framework, we (i) explore a semi-synthetic setting where quality can be clearly defined and controlled, and (ii) move beyond the limitations of earlier experiments, such as fixed model size and finite training horizon, which only offer snapshots of the following analysis.

### 6.1 DATA CONDITIONING: DATA-QUALITY AS AN OPTIMIZATION CATALYST

Central to our analysis is the concept of data conditioning, which we define as a desirable property of data quality. Informally, a dataset $D_{\text{clean}}$ is better data conditioned than another dataset $D_{\text{dirty}}$ if a model trained on $D_{\text{clean}}$ outperforms a model trained on $D_{\text{dirty}}$ when evaluated on $D_{\text{dirty}}$.

We describe it formally as follows. Given an objective function $\ell$ and a dataset $D$, we define the loss function as $\mathcal{L}(\theta, D) := \mathbb{E}_{x \sim D}[\ell(\boldsymbol{x}; \theta)]$. This loss is typically approximately minimized by running a stochastic optimization algorithm $\mathcal{A}$ on the samples $\boldsymbol{x}_i$:

$$\theta_{D_{\text{dirty}}}^n \leftarrow \mathcal{A}(\boldsymbol{x}_i), \text{ with } (\boldsymbol{x}_i)_{i=1}^n \sim D_{\text{dirty}}, \tag{2}$$

where $\boldsymbol{x}_i$'s are $n$ i.i.d. samples from $D_{\text{dirty}}$. Instead of training on $D_{\text{dirty}}$, one can also train on $D_{\text{clean}}$ and obtain parameters $\theta_{D_{\text{clean}}}^n$. We propose an axiomatic definition of quality:

> **Data-conditioning.** We write $D_{\text{clean}} \succ D_{\text{dirty}}$ and say that a dataset $D_{\text{clean}}$ is better data-conditioned than $D_{\text{dirty}}$, relative to the learning rule $\mathcal{A}$ and the horizon $n \in \mathbb{N}$ if
>
> $$\mathcal{L}(\theta_{D_{\text{clean}}}^n, D_{\text{dirty}}) \leq \mathcal{L}(\theta_{D_{\text{dirty}}}^n, D_{\text{dirty}}). \tag{3}$$

We coin this phenomenon "data conditioning", drawing from the optimization literature, where *conditioning* typically describes how easily a loss function can be minimized. In our context, data conditioning captures how the structure of a dataset accelerates optimization. Indeed, in standard large-scale settings, data are seldom repeated, and models generalize well, which means that the training loss closely approximates the validation loss. Therefore, if we had a perfect minimization oracle, $\mathcal{A}(\boldsymbol{x}_i) = \arg\min_{\theta \in \Theta} \frac{1}{n} \sum_{i=1}^n \ell(\boldsymbol{x}_i, \theta)$, we would have by definition of the minimizer $\mathcal{L}(\theta_{D_{\text{dirty}}}^n, D_{\text{dirty}}) \simeq \frac{1}{n} \sum_{i=1}^n \ell(\boldsymbol{x}_i, \theta_{D_{\text{dirty}}}^n) \leq \frac{1}{n} \sum_{i=1}^n \ell(\boldsymbol{x}_i, \theta_{D_{\text{clean}}}^n) \simeq \mathcal{L}(\theta_{D_{\text{clean}}}^n, D_{\text{dirty}})$. This would forbid the existence of better data-conditioned datasets. However, the existence of better-conditioned datasets has been reported many times in the literature, and is at the root of curriculum

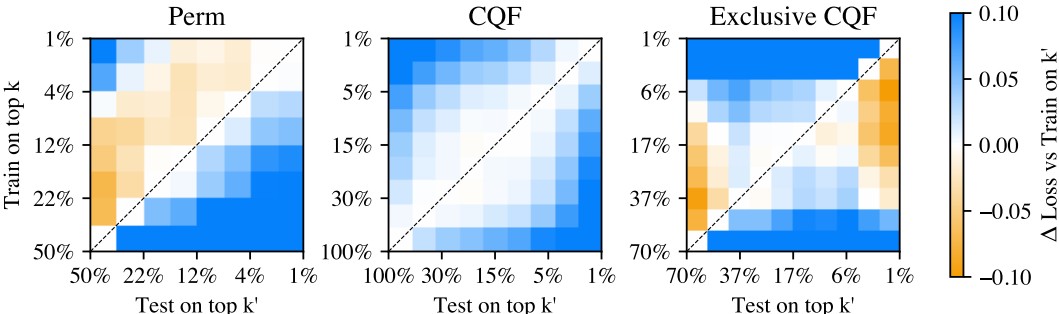

Figure 8: **Data conditioning experiment.** We use three different ways to define an axis of "quality", which are datasets indexed by a scalar $k \in [0,1]$, where $k = 0$ means higher quality. Perm defines it as $(1 - k)$ where $k$ is the probability of randomly permuting a document. CQF defines it as the fraction of documents kept in the pretraining set, where the HQ set is OpenOrca. Exclusive CQF defines it as documents that have scores between two thresholds. Each of these datasets is parameterized by a quality knob, $k$. We train models for a grid of values $k$, and compute their test loss on the dataset $k'$, $\mathcal{L}(k, k')$. The figure displays the matrices with entries $\mathcal{L}(k, k') - \mathcal{L}(k', k')$. A negative value for the coefficient $k, k'$ means $k \succ k'$, as defined in Equation 3.

learning (Bengio et al., 2009), dataset distillation (Wang et al., 2018), or mixture optimization (Zhang et al., 2025; Shukor et al., 2025). Thus, our definition of quality arises from imperfect optimization.

We believe that data-conditioning can act as a guiding principle for data filtering. Indeed, if one has two datasets such that $D_{\text{clean}} \succ D_{\text{dirty}}$, there is no use in training on $D_{\text{dirty}}$, if we have enough tokens in $D_{\text{clean}}$, because it would yield an inferior model even on the distribution it is trained on. This can therefore be seen as a data-selection principle: how can we select a subset in $D_{\text{dirty}}$ that is better data-conditioned than $D_{\text{dirty}}$ itself?

### 6.2 CQF THROUGH THE LENS OF DATA-CONDITIONING

To illustrate this notion of data conditioning, we explore different ways of creating a spectrum of "quality", using families of datasets indexed by one variable $k \in [0,1]$, where, intuitively, lower values of $k$ correspond to higher-quality datasets, and higher values indicate lower-quality ones.

First, we create semi-synthetic text datasets with varying levels of quality, inspired by Kallini et al. (2024). Using RedPajama-V2 as our base dataset representing the highest quality, we simulate different quality levels by constructing a family of datasets $\text{Perm}(k)$ for $k \in [0,1]$. Each $\text{Perm}(k)$ is created by sampling documents whose tokens are randomly permuted with probability $k$, or kept unchanged with probability $1 - k$. Similarly, we define another family of datasets $\text{CQF}(k)$, where $k$ denotes the selection fraction, using CQF with OpenOrca as the HQ set. We define Exclusive CQF by taking documents whose score lies in a given interval. We compare the scaling behaviors of models trained on each of these datasets, by varying the number of parameters $N$, training tokens $D$, and the "quality" level $k$. We report the next-token prediction loss of these models on each "quality" level $k'$.

**Static analysis.** We begin by training models of a fixed size for a fixed number of iterations on each quality bucket $k$ in Figure 8. Each index $(k, k')$ shows the value $\mathcal{L}(k, k') - \mathcal{L}(k', k')$, where $L(k, k')$ is the loss on quality bucket $k'$ for a model trained on quality bucket $k$. For the synthetic case, we observe a mostly upper-triangular structure, which means that training on better quality domains also improves models on lower quality domains, apart from the edge case of training on non-permuted tokens. In other words, organizing data by quality deciles leads to structured performance gains in this controlled setting, where higher-quality data results in greater improvements, aligning with our intuition of quality as a concept. In Appendix E we extend our investigation of this binary relation.

**How does data conditioning depend on scale?** For the $\text{Perm}$ quality axis, we repeat the previous experiment at different model scales and training horizons, with model scales ranging from 125M to 1.3B parameters. Then, for each train/validation pair $k, k'$, we fit a scaling law that predicts the loss $\mathcal{L}(k, k')$ as a function of $N$, the model size, and $D$, the number of seen tokens. We fit the Chinchilla

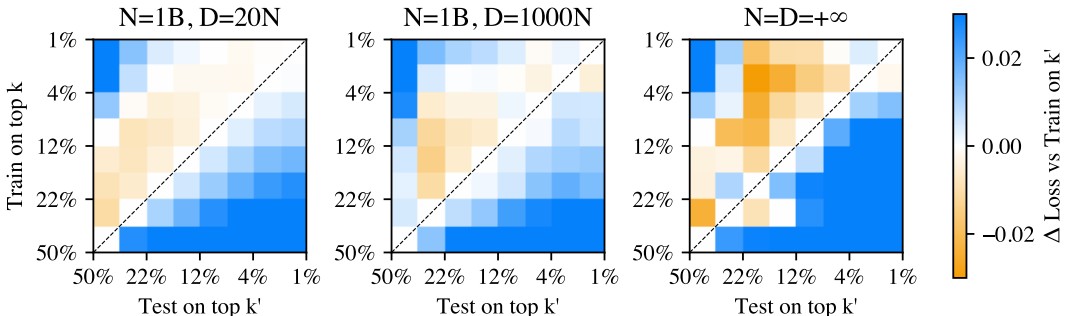

Figure 9: **Data conditioning scaling.** We fit scaling laws in order to have a dynamic view of Figure 8 (left). We then report the predicted loss of models of size $N$ trained with $D$ tokens. When $N = D = +\infty$, we use the irreducible error term $E$ predicted by the scaling law as a proxy for the loss. We observe that the regions of better data-conditioning (orange) are mostly kept the same as we scale models. When scaling in the large D direction, we observe that the effect gets narrower.

scaling law (Hoffmann et al., 2022):

$$L(k, k')_{N,D} = E + \frac{A}{N^\alpha} + \frac{B}{D^\beta}$$

where the parameters $E, A, B, \alpha, \beta$ depend on the train/validation pairs $k, k'$. This enables us to obtain a dynamic version of Figure 8 in 9, where the model sizes and number of tokens are variable. These findings validate that data-conditioning is only mildly dependent on the model and data scale. It means that data-conditioning can be validated through small-scale proxy models, and then leveraged with large-scale models.

## CONCLUSION

Classifier-based Quality Filtering is a tool used to train most state-of-the-art models, yet our analysis shows that its inner workings are more subtle than previously believed. While CQF reliably improves downstream evaluations, these gains are not attributable to the fact that filtered data are closer to the high-quality set. Instead, we uncover an implicit filtering phenomenon, where CQF emphasizes HQ examples that are far from the bulk of the LQ set, and are therefore most likely to be of higher quality. Quality filtering is about removing the "bad", not imitating the "good".

Finally, we challenge the notion of quality defined by CQF, demonstrating that it does not satisfy the desirable property of *data conditioning*: training on "better quality" data, according to CQF, does not accelerate learning on lower quality subsets. CQF should not be taken as a universal quality measure, but instead as a way to better align with downstream evaluations.

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

CONTENTS

# A APPENDIX ORGANIZATION

The appendix is organized as follows:

- In Appendix B, we study how the optimal fraction $k$ of selected data in CQF varies with model size and training compute, the HQ set and the downstream task.

- In Appendix C, we highlight that CQF classifiers are prone to learning spurious features, such as context length, and we evaluate the effectiveness of a simple mitigation strategy. This illustrates a broader phenomenon: CQF can induce undesired biases that cause the selected pretraining data to diverge significantly from the HQ set.

- In Appendix D, we reveal that no single HQ set leads to universally better downstream performance, and that different classifiers implicitly align with different benchmarks, revealing task-specific inductive biases.

- In Appendix E, we visualize the binary relation induced quality filtering as a graph, highlighting how its structure evolves from the semi-synthetic setting to CQF used in practice.

- In **??**, we provide the reader with further implementation details.

# B OPTIMAL THRESHOLDS VARY WITH COMPUTE

How to chose the optimal $k$ when picking the top $k\%$ documents from CQF? To answer this, we conducted a series of ablations over $k$, training models on the top $k\%$ of the pretraining data, as ranked by CQF, using various HQ sets. These experiments span multiple model sizes $N$ and training horizons $D$ (i.e., number of seen tokens), such that the total training compute in FLOPs is measured as $6ND$. The results are summarized in Figure 10, where we report downstream accuracy as a function of training FLOPs and highlight the optimal $k$ in each setting.

Although our setup directly illustrates CQF, making it more representative of real-world data filtering pipelines, Mizrahi et al. (2025) concurrently explore a related direction. Their approach differs in that they select LQ data based on direct proximity to target benchmarks, bypassing the need for a proxy HQ dataset. Despite this, our findings do not align: we observe no clear trend once the noise level is accounted for, leading to relatively inconclusive results. We also note that Mizrahi et al. (2025)'s conclusions rely on extrapolation, which probably explains the divergence.

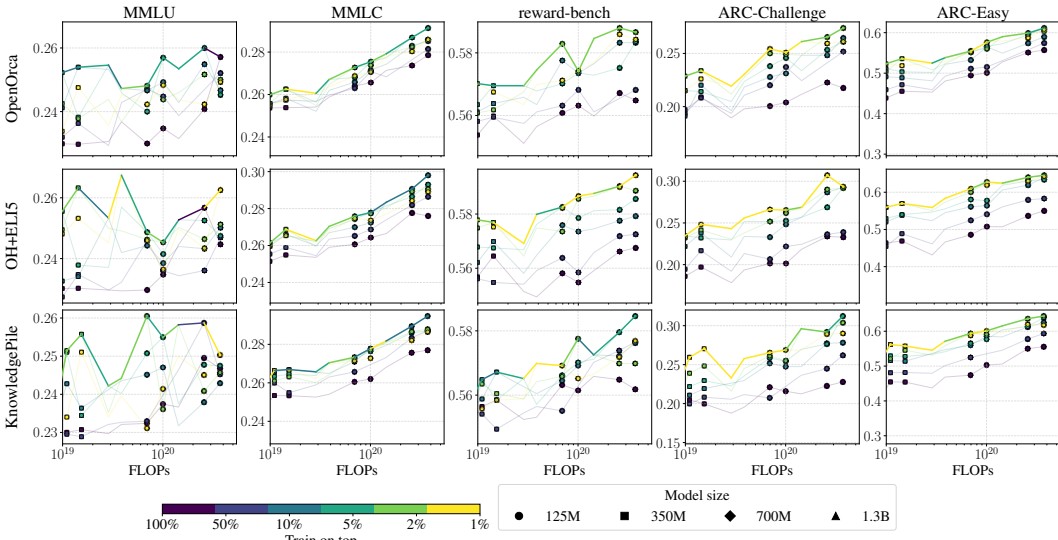

Figure 10: **The optimal top $k\%$ of pretraining data depends on available compute.** For each setting, we highlight the value of $k$ that yields the best performance under a fixed compute budget. **Rows**: different HQ sets used for CQF. **Columns**: various downstream performance metrics.

## C  DO CLASSIFIERS USED IN CQF EXHIBIT UNDESIRED BIASES?

Even when downstream performance improves, the selected data can drift from the intended target distribution—revealing not only a failure to capture genuine quality, but also an undesirable inductive bias, where the classifier overemphasizes unrelated features.

Whilst it is not trivial to exhibit such unwanted features among the learned ones by the classifier, we managed to identify one of these for OpenOrca as HQ set: the classifier seems to associate quality with the sequence length, and shorter sentences have higher chances to be classified as high quality ones, see Figure 11.

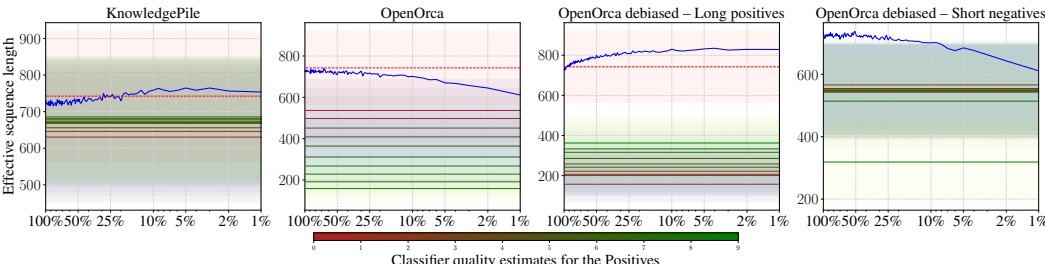

Figure 11: **CQF classifiers suffer from inductive biases.** Because the OpenOrca dataset (HQ set) contains shorter sequences than RedPajama (LQ set), the classifier in CQF learns to use sequence length as proxy for quality scores (**second column**). This bias persists even after filtering out long documents from OpenOrca (**third column**), and only disappears when we subsample the negative class to match shorter sequence lengths (**fourth column**). In contrast, the classifier from CQF using KnowledgePile as a HQ set (**first column**) does not exhibit this behavior. The red dotted line indicate the effective sequence length in the HQ set, while the blue line shows the sequence length of data filtered by CQF at different selection ratios along the x-axis. The HQ set is divided into 10 quality deciles, and the sequence lengths for each decile are shown as solid horizontal lines, with color indicating quality level.

When sampling from the positive class (OpenOrca dataset) prior to training the corresponding classifier, we subsample documents with an imposed sequence length of at least 500 or 700. We then use this classifier to produce a partition of RedPajama with an updated notion of quality, that we hope to be seemingly better or at least not mistakenly taking sequence as a proxy for quality; see columns 3 and 4 of Figure 11. We train 350M models on the resulting partitions of RedPajama and evaluate them on ARC (Clark et al., 2018), MMLU (Hendrycks et al., 2021), and Reward Bench (Lambert et al., 2025). We show in Figure 12 the result of such experiments, averaged across 3 runs.

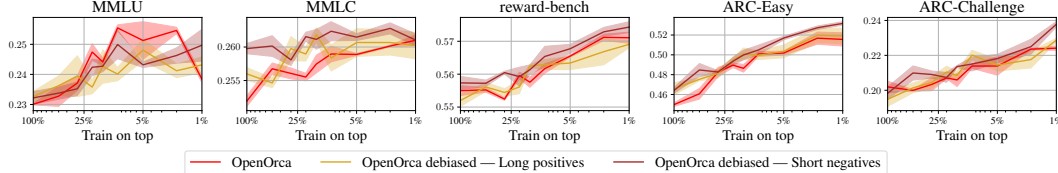

Figure 12: **Performance after debiasing the classifier from CQF with OpenOrca as a HQ set.** The classifier was retrained with a subsampled HQ set (OpenOrca) using minimum sequence lengths, in an effort to remove length-based bias in quality scores.

Beyond this specific case of sequence length bias, we investigate whether CQF classifiers exhibit similar issues, when trained on HQ sets drawn directly from target benchmarks. To assess this, we compute sBert embeddings for RedPajama documents grouped by CQF quality scores and compare them to embeddings of the benchmark data. As shown in Figure 13, we visualize the centroids of each quality bucket using a two-dimensional UMAP projection. Ideally, higher-quality buckets as ranked by CQF (darker colors) would be closer to the benchmark embeddings. We provide the same visualization in Figure 14 using a PCA. Surprisingly, this is often not the case, suggesting that classifiers may still rely on spurious correlations or unrepresentative features of the entire HQ set.

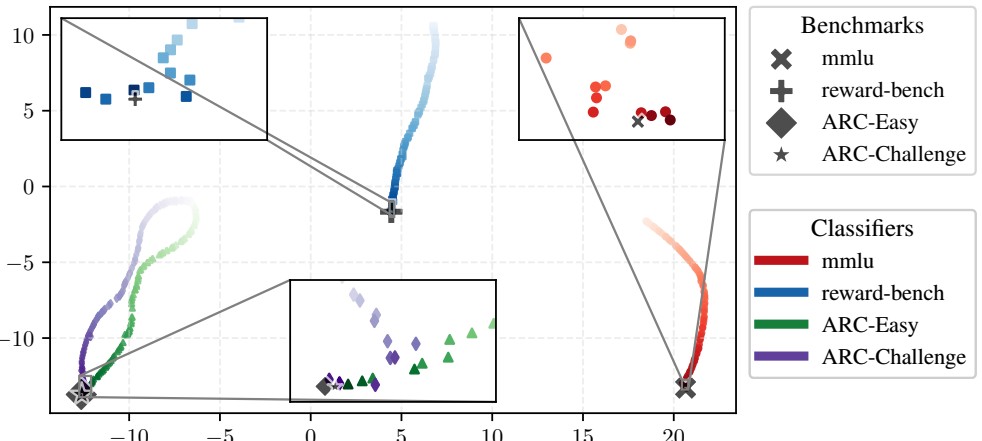

Figure 13: **UMAP of sBert centroids for each (exclusive) quality bucket.** Even when quality classifiers are trained directly on the target data, they may still capture undesirable features. Consequently, the top-rated RedPajama quality buckets (darker colors) are not always the closest to the target benchmark embeddings.

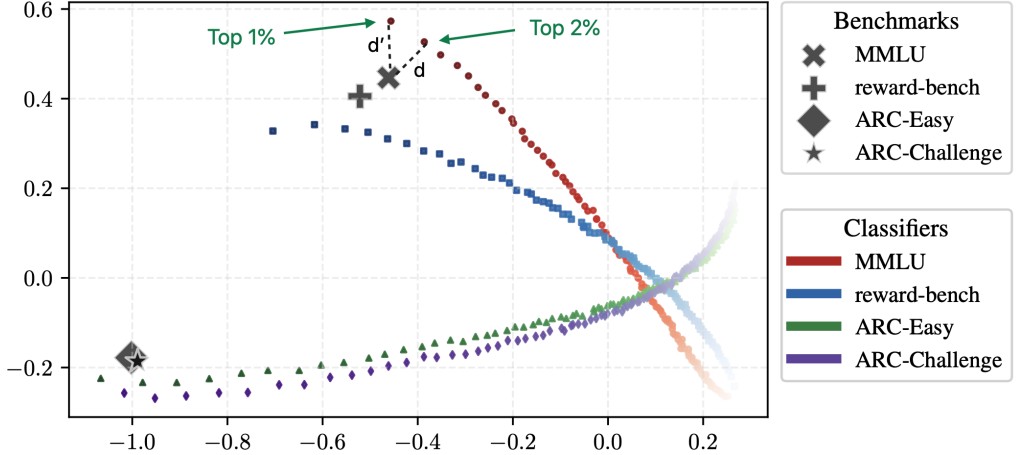

Figure 14: **PCA of sBert embeddings of (exclusive) quality buckets induced by different classifiers.** Even when quality classifiers are trained directly on the target downstream tasks, they may still capture undesirable features. Consequently, the top-rated RedPajama quality buckets (darker colors) are not always the closest to the target benchmark embeddings.

Finally, we provide a 2D visualization of the sBert latent space using a tSNE from which similar conclusions can be drawn in that only a subset of the HQ set is matched by the data retained from CQF.

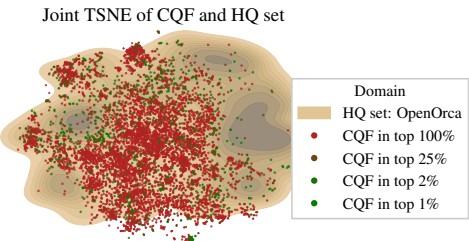

Figure 15: **2D TSNE of sBert embeddings of OpenOrca and CQF samples**. The TSNE reveals the same insights as the 2D PCA in Figure 4. This method also shades lights on the difficulty of properly projecting and representing in 2D a 384-dim geometry.

## D  NO HQ SET IS SUPERIOR TO ALL OTHERS ACROSS ALL TASKS

While various HQ sets are used in the literature for CQF, no single HQ consistently outperforms others across all downstream tasks. Figure 16 shows that varying HQ sets yield various performance across tasks, with no universal dominance. Downstream evaluations are noisy, but we observe the consistent trend that OH+ELI5 is a good baseline across tasks, confirming the findings of Li et al. (2024). We also notice that KnowledgePile, despite poor diversity in the style, induce a bias toward data is are more heavily leaning toward knowledge benchmarks like ARC.

This suggests that each HQ set imparts its own inductive biases, influencing which aspects of the data are emphasized during filtering. To further understand these biases, we visualize the embedding space of the data selected by each classifier in Figure 18. We observe that quality buckets across classifiers tend to align with specific benchmark datasets, indicating that classifiers—implicitly or explicitly—favor data that resembles their respective supervision targets. This aligns with recent concurrent work from Mizrahi et al. (2025), who show that direct supervision using explicitly target benchmark data can boost performance on that benchmark, though at the cost of generality. Taken together, these results highlight a central challenge in CQF: quality is not a universal property, and each HQ set carries task-specific preferences that limit its transferability.

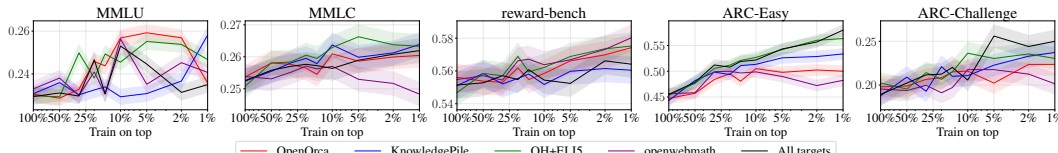

Figure 16: **Benchmark performance results from 350M models** trained on documents ranked by quality according to various CQF using various HQ sets.

All the manifold visualizations in Figure 17 and Figure 18 demonstrate the same trend: CQF selects data closer to benchmarks as quality filtering goes.

## E  DATA CONDITIONING

We revisit the experiments of Figure 8 by materializing the graph induced by the binary relation $\succ$. For an arbitrary algorithm $\mathcal{A}$ it is hard to characterize the datasets $D_{\text{clean}}$ and $D_{\text{dirty}}$. Therefore, we rely on empirical measurements draw edges when the loss improvement is significant (e.g. bigger than the standard deviation). The results are given in figs. 19 and 20.

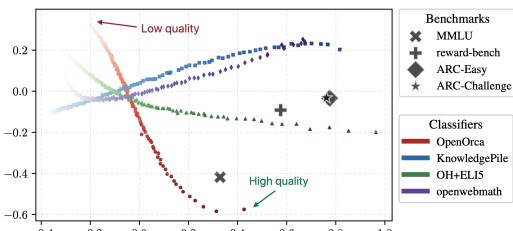

Figure 17: **PCA embedding of (exclusive) buckets.** This figure differs from Figure 3 by considering exclusive buckets. Here, we see that the bottom 10% are quite different from each other, and the buckets of average quality (i.e in the 70-30 range) tend to be similar across quality classifiers.

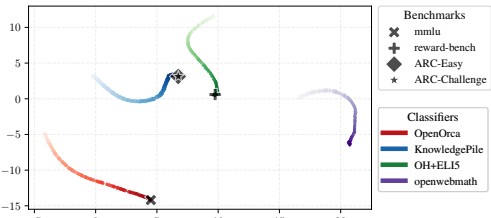

Figure 18: **Each HQ set used in CQF appears to favor task-specific data.** Two-dimensional UMAP of sBert centroids for each (exclusive) quality bucket as defined by each classifier. Darker color indicates increasing selection ratio $k$.

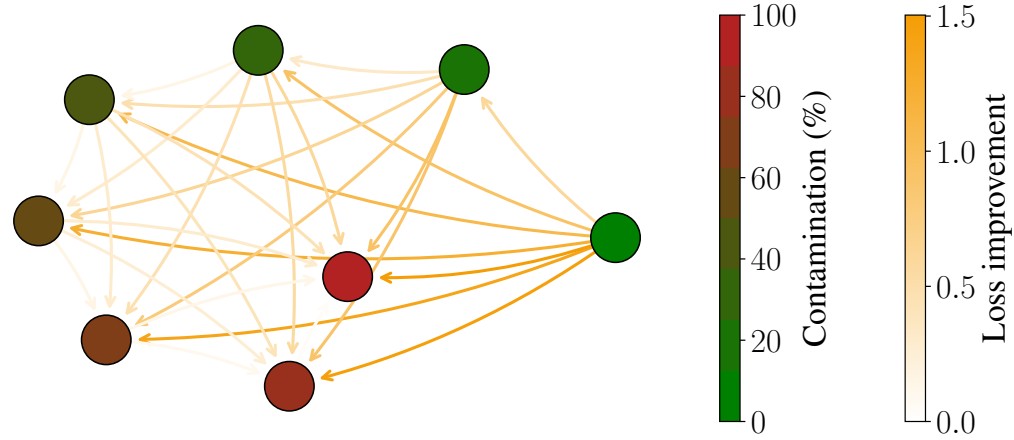

Figure 19: **Data conditioning $\succ$ on the Perm task.** This graph exhibits the properties of a total ordering, closer to an intuitive notion of quality. The only "backward" edge is linking the the two worse splits, and the loss difference is within standard deviation.

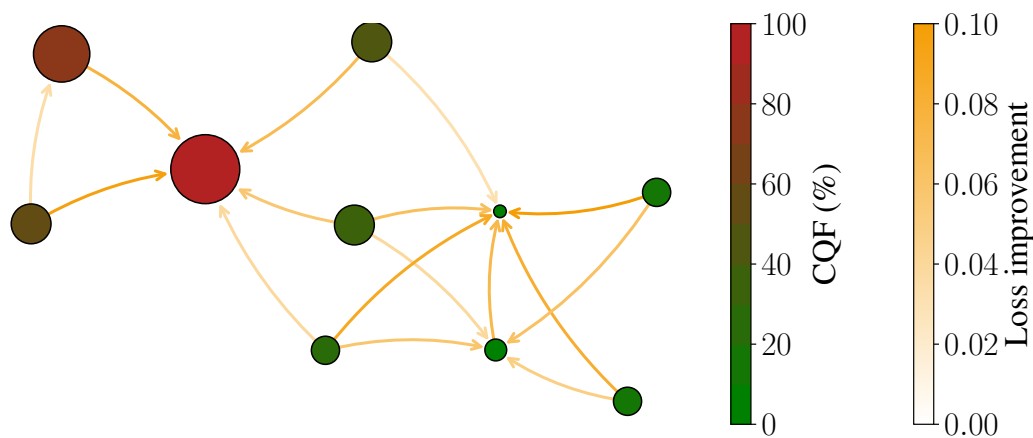

Figure 20: **Data conditioning ≻ on OpenOrca CQF.** On these exclusive buckets, there is no global ordering. The bottom 30% (red) and the top 5% (green) are dominated by bucket of "average" quality (possibly with more diversity). The node size is proportional to the number of examples in the bucket. On this graph, the relation is transitive, which induces an ordering, but this ordering is not total.

