# OpenReview forum: "The data-quality illusion: Rethinking Classifier-based Quality Filtering for LLM Pretraining"
_ICLR.cc/2026/Conference — Submitted to ICLR 2026_

### Official Review · Reviewer_5AvM · 2025-10-29

**Soundness:** 4
**Presentation:** 3
**Contribution:** 3
**Rating:** 6
**Confidence:** 4

**Summary:**

The paper critically examines classifier-based quality filtering, a widely used method for pretraining data selection. It shows that CQF improves downstream task performance but paradoxically doesn’t improve language modeling on the high-quality data. The authors argue that CQF implicitly filters the HQ set itself and captures stylistic similarity rather than true data quality. They introduce "data conditioning" as a new lens to evaluate whether filtering improves optimization dynamics.

**Strengths:**

- Excellent empirical and theoretical dissection of a widely used but poorly understood technique.
- Timely and relevant analysis of a core assumption in LLM data engineering.
- Clear exposition of the paradox and insightful explanation linking CQF to implicit HQ filtering.
- Strong experimental design and visualizations that clarify complex effects.

**Weaknesses:**

- This work is largely diagnostic with limited actionable guidance on improving CQF beyond critique.
- Data conditioning concept, while elegant, remains somewhat abstract and untested in real large-scale settings.
- All experiments are conducted at modest scale which may not generalize.
- Relies heavily on pretraining proxies (ARC, MMLU) rather than practical LLM evaluation.

**Questions:**

- Can the proposed data conditioning principle be used to design better filtering methods?
- How robust are the findings when using multilingual datasets?
- Does the illusion persist when HQ sets are human-curated instruction data versus web data?
- Could the implicit HQ filtering effect be leveraged deliberately (e.g., via adaptive weighting)?

Missing citations:
- When Less is More: Investigating Data Pruning for Pretraining LLMs at Scale, Max Marion, Ahmet Üstün, Luiza Pozzobon, Alex Wang, Marzieh Fadaee, Sara Hooker
- Deep Ignorance: Filtering Pretraining Data Builds Tamper-Resistant Safeguards into Open-Weight LLMs, Kyle O'Brien, Stephen Casper, Quentin Anthony, Tomek Korbak, Robert Kirk, Xander Davies, Ishan Mishra, Geoffrey Irving, Yarin Gal, Stella Biderman

---

> ### Author Response · Authors · 2025-11-24
>
> Dear reviewer,
>
> We thank you for your review and for taking time to read our paper. We are happy to hear that you found our analysis “insightful” and supported by “excellent experimental dissection”.
>
> We now answer the questions you have raised.
>
> > This work is largely diagnostic with limited actionable guidance on improving CQF beyond critique.
>
> We believe that providing a detailed analysis of a method widely used for data filtering is in itself a valuable contribution. We refer the reviewer to our general response, where we clarify in detail the practical implications of this work. Indeed, our findings challenge the widespread assumption that simply adding more HQ data is always the best course of action, which offers a clear actionable guidance on current data processing pipelines.
>
> > Data conditioning concept, while elegant, remains somewhat abstract and untested in real large-scale settings.
>
> We evaluate this notion for several model scales, on practical datasets. Please see our general response, where we detail how our definition of data conditioning can be effectively implemented using small-scale proxies.
>
> > All experiments are conducted at modest scale which may not generalize.
>
> We are currently running experiments at 3B and 7B scale. We will share the results upon completion.
>
> > Relies heavily on pretraining proxies (ARC, MMLU) rather than practical LLM evaluation.
>
> These downstream evaluations are usually regarded by the community as a gold standard for assessing pretraining performance. Evaluations targeting later stages of the LLM pipeline, such as instruction following or reasoning capabilities, falls beyond the scope of this paper which solely tackles pretraining dynamics, rather than post-training behavior for which such evaluations would make much more sense.
>
> > Can the proposed data conditioning principle be used to design better filtering methods?
>
> While it is proposed as a theoretical principle, we believe that it could also be used as a diagnostic tool for designing training datasets: indeed, if A is better data conditioned than B, there is no use in training models on B (unless data in A is very limited); and this can be measured easily with proxy models. Indeed our scaling law analysis shows that the trend is quite independent of model scale. This can therefore be used as a guiding principle for practitioners. Developing data filtering methods based on this principle \- such as: given a dataset A, how to extract a subset B such that B \> A? \-  is an interesting future research direction. We have clarified this in the text.
>
> > How robust are the findings when using multilingual datasets?
>
> Redpajama-v2 is already a multilingual dataset, even though the HQ sets are English only. Investigating how CQF behaves across languages within the HQ set would be an interesting direction for future work—thank you for highlighting this.
>
> > Does the illusion persist when HQ sets are human-curated instruction data versus web data?
>
> The datasets used, such as openorca or openhermes, are close to being human-curated instruction data. ELI5 is actually  human-generated, as it comes from a Reddit thread. Openwebmath and ARC Easy are human-generated datasets. Therefore, yes, the illusion persists.
>
> > Could the implicit HQ filtering effect be leveraged deliberately (e.g., via adaptive weighting)?
>
> One of the messages of the paper is that it is already leveraged; this implicit filtering is the reason why CQF helps improving downstream tasks while at the same time leading to worse language modeling on the HQ set as k decreases.

---

### Official Review · Reviewer_dHu5 · 2025-11-01

**Soundness:** 3
**Presentation:** 2
**Contribution:** 2
**Rating:** 4
**Confidence:** 2

**Summary:**

The paper argues that the perceived benefits of Classifier-based Quality Filtering (CQF) in large language model (LLM) pretraining are largely illusory.

Although CQF improves downstream benchmark performance, it does not actually select data that better resemble “high-quality” corpora, nor does it improve language modeling performance on such data.

Instead, CQF works primarily by removing obvious low-quality samples and by reweighting the pretraining distribution toward benchmark-style text.

**Strengths:**

1. Elegant theoretical framing (density-ratio view) that unifies prior empirical quirks.

2. Strong empirical design across multiple HQ datasets.

3. Clear visualizations demonstrating domain drift.

**Weaknesses:**

1. The filtering process also reduces token count; loss differences may stem from fewer tokens, not “quality”.

2. Circular validation: The “HQ-decile” analysis (Fig. 5) reuses the same CQF score to both partition and evaluate HQ samples, effectively validating the classifier with itself rather than demonstrating genuine quality correlation.


2. The scaling-law fitting is not convincing.

    - The experiments vary both data size and distribution with k, violating the assumption of a fixed task distribution.

    - Only three data points (1 %, 10 %, 100 %) are used, with uncontrolled compute budgets, making the fitted β unreliable.

    - No residuals or confidence intervals are reported.

I recommend removing or reframing this section; at present, it does not provide meaningful evidence of scaling behavior.

**Questions:**

see weaknesses.

---

> ### Author Response · Authors · 2025-11-24
>
> Dear reviewer,
>
> We thank you for your review and for taking time to read our paper. We are happy to hear that you found our analysis “elegant” and supported by “strong empirical design”.
>
> We now answer the questions you have raised.
>
> > The filtering process also reduces token count; loss differences may stem from fewer tokens, not “quality”.
>
> We want to clarify a misunderstanding: as explicitly stated in the text (lines 196-198), “One clear limitation of CQF is that the number of training tokens available in the dataset is k × D where D is the total number of tokens in the LQ set. Too small values of k lead to scarce datasets on which models cannot be trained without repeating data or even overfitting. In this paper, we step away from this limitation and always use values of k such that there are enough data in DCQF to train a model without repeating data.”: all the training runs in the paper have been done **without repeating data**. CQF is never limited by the number of training tokens, and all runs are done with the same number of tokens regardless of k.
>
> > Circular validation: The “HQ-decile” analysis (Fig. 5) reuses the same CQF score to both partition and evaluate HQ samples, effectively validating the classifier with itself rather than demonstrating genuine quality correlation.
>
> We are unsure what you mean by this comment, how is it a circular validation? Fig 5 shows that models trained with a small k perform well on the HQ set with a high score, effectively showing an implicit partitioning of the HQ set as well. The goal of the figure is not to demonstrate a “genuine quality correlation”, but to understand the inner workings of CQF.
>
> > The scaling-law fitting is not convincing. The experiments vary both data size and distribution with k, violating the assumption of a fixed task distribution.
>
> We want to clarify that data size (D) is usually changed when fitting scaling laws. We fit one scaling law per task distribution (one for each value of k and k’), which means that the fixed task distribution is not violated.
>
> > Only three data points (1 %, 10 %, 100 %) are used, with uncontrolled compute budgets, making the fitted β unreliable.
>
> This is a hallucination on the reviewer's part. We never mentioned using 3 data points. As written in the text, and seen in the axes of fig8 and 9, we do not use such a small grid. We are unsure what the reviewer means by uncontrolled compute budget, as we clearly control the values of N, D used to train each model.
>
> > No residuals or confidence intervals are reported
>
> We will include the scaling laws Mean Residual Error on held-out data in the text. We have one value for each (k, k’) train-test pair; it is generally <1%, which indicates a good scaling law fit.
>
> > I recommend removing or reframing this section; at present, it does not provide meaningful evidence of scaling behavior.
>
> We disagree with the reviewer, our scaling law section shows the negligible effect of both tokens and model size on our proposed definition of data quality. This finding effectively shows that small-scale experiments can be used as reliable good proxies for assessing the quality of a dataset A over B according to our definition. This section makes our definition usable in practice.

---

### Official Review · Reviewer_vK32 · 2025-11-01

**Soundness:** 2
**Presentation:** 3
**Contribution:** 2
**Rating:** 4
**Confidence:** 4

**Summary:**

This paper revisits Classifier-based Quality Filtering (CQF) — a standard data curation method in LLM pretraining. The authors find a paradox: CQF boosts downstream task performance but doesn’t improve language modeling on the high-quality (HQ) dataset. They explain this by showing CQF implicitly reweights the HQ set itself, emphasizing samples far from the low-quality (LQ) data. They also contrast CQF with importance sampling and introduce data conditioning as a new, optimization-based notion of data quality.

**Strengths:**

- The writing is clear and easy to follow.


- The paper identifies a previously overlooked paradox in CQF, showing that it may not improve language modeling on high-quality data even when it enhances downstream task performance — a perspective that challenges long-standing assumptions in data filtering.


- Introducing data conditioning as an optimization-based definition of data quality represents a significant conceptual advance, offering a new lens to evaluate dataset usefulness beyond static classifier scores.

**Weaknesses:**

- While the data conditioning notion is conceptually sound but too ideal, it is impractical for real-world data filtering — evaluating it requires repeated model training and loss computation across datasets, which is computationally expensive and cannot be efficiently estimated per sample. Thus, it serves more as a diagnostic concept rather than a usable filtering metric. From this perspective, the original CQF-style “remove-the-bad” metric can be viewed as a more conservative yet pragmatic strategy for large-scale data curation. Moreover, this paper fail to provide an alternative practical metric to


- The experimental analysis, while extensive, is conducted on relatively limited model scales (≤1.3B) and specific datasets (RedPajama-V2, OpenOrca, KnowledgePile). It is unclear whether the same paradoxical behavior of CQF would persist under larger-scale models (such as 7B) or more diverse corpora with heterogeneous noise patterns.

**Questions:**

1. The data conditioning notion is theoretically appealing but computationally impractical. How could it be approximated or operationalized for large-scale data filtering — e.g., via small-model proxies, gradient statistics, or early-training dynamics?

2. The paper analyzes how the CQF selection fraction (k) affects downstream and HQ-set performance, showing that smaller k values emphasize samples farther from the LQ distribution. However, the analysis does not explicitly quantify how the relative size of the selected subset influences this implicit reweighting. When k becomes large—approaching the size of the HQ set itself—does CQF recover most HQ-like data, or does it still inevitably include LQ-like regions? Could the authors clarify how the composition (HQ-aligned vs LQ-aligned) of the filtered dataset evolves with k and whether this proportion can be estimated empirically?

---

> ### Author Response · Authors · 2025-11-12
> **Missing end of line**
>
> Dear reviewer,
>
> Thank you very much for your comments, which we will address promptly. There seems to be a missing end of line for the first weakness that you raise: *"Moreover, this paper fail to provide an alternative practical metric to"* . Could you please expand on what you mean?
>
> Best regards,
> The authors.

---

> > ### Comment · Reviewer_vK32 · 2025-11-12
> >
> > It should be "Moreover, this paper fails to provide a practical alternative metric to operationalize data conditioning, leaving a noticeable gap between the theory and its real-world applicability."

---

> ### Author Response · Authors · 2025-11-24
>
> Dear reviewer,
>
> We thank you for your review and for taking time to read our paper. We are happy to hear that you found our analysis “represents a significant conceptual advance” and that you thought the paper was well written.
>
> We now answer the questions you have raised.
>
> > While the data conditioning notion is conceptually sound but too ideal, it is impractical for real-world data filtering — evaluating it requires repeated model training and loss computation across datasets, which is computationally expensive and cannot be efficiently estimated per sample.
>
> Indeed, this notion requires training models. However as the scaling law results indicate, this notion is relatively scale agnostic, which means that it can be estimated from small scale proxy runs. We have expanded on this in the main text. We agree with the reviewer that this definition does not yield a per-sample estimate of data quality, as it is inherently defined at the dataset level since it takes into account the optimization trajectory (i.e., how well the objective is minimized). Further works might consider sample-level variants perhaps by considering gradients, which would be slightly orthogonal to the present. Thanks for raising this interesting point\!
>
> > From this perspective, the original CQF-style “remove-the-bad” metric can be viewed as a more conservative yet pragmatic strategy for large-scale data curation.
>
> Indeed, this is what we argue in fig.4. We still believe that the analysis in sec. 4 is required to fully understand CQF, as a method biased towards data that is far from the LQ set. Moreover, please note that, CQF also requires multiple training runs in practice to tune the selective fraction k as it is unclear how well this parameter scales with model size and tokens. The reviewer might find our exploration interesting in Appendix B, where we precisely investigated possible trends for the best value of k across scale and found the results inconclusive, suggesting that this parameter would need to be tailored to specific problems through multiple training runs.
>
> >  It is unclear whether the same paradoxical behavior of CQF would persist under larger-scale models (such as 7B) or more diverse corpora with heterogeneous noise patterns.
>
> RedPajama v2 is already a very diverse corpora, including over 32 T tokens from diverse web sources (multilingual, math, code, etc.). Regarding larger-scale runs, we have included results from training models from scratch up to 1.3B parameters. We are currently working on scaling these experiments further and will try our best to provide additional results before the deadline.
>
> > The data conditioning notion is theoretically appealing but computationally impractical. How could it be approximated or operationalized for large-scale data filtering — e.g., via small-model proxies, gradient statistics, or early-training dynamics?
>
> As you mention, a first answer lies in small-model proxies: as illustrated in fig 9, this notion is quite orthogonal to scale. A pragmatic approach to evaluate data-conditioning is therefore to estimate it with small scale models. We made this important fact clearer in the text. Further analysis regarding gradient dynamics would be an interesting future research direction, especially looking at the bias/variance trade-off of gradients. Thanks for raising this point!

---

> ### Author Response · Authors · 2025-11-24
>
> > However, the analysis does not explicitly quantify how the relative size of the selected subset influences this implicit reweighting. When k becomes large—approaching the size of the HQ set itself—does CQF recover most HQ-like data, or does it still inevitably include LQ-like regions?
>
> We want to clarify that CQF acts as a binary selection over the LQ set: the CQF set is a subset of the LQ set, hence it contains LQ-like data. When k is small, CQF draws from a small fraction of the LQ set, and when k is large, it spans the entire LQ set. In all cases, we subsample from the resulting CQF set to ensure an equal token count across training runs. Additionally, as stated in lines 196-198: “*In this paper, we \[...\] always use values of k such that there are enough data in $D\_{CQF}$ to train a model without repeating data. This allows us to focus solely on the impact of data quality rather than on the effects of repeated training examples.*”
> Therefore, we have never explored values of k that yield a CQF set as small as the HQ set, since it would entail having too few samples (because the HQ set is very small) and would prevent training without repetitions, as required in all experiments of this paper.
> In any case, given the results in fig.4 where the loss on the HQ set is not monotonous with k, we can conclude that CQF does not select data from the LQ set that is systematically close to the HQ set.
>
> > Could the authors clarify how the composition (HQ-aligned vs LQ-aligned) of the filtered dataset evolves with k and whether this proportion can be estimated empirically?
>
> This is a good point. We have added a row in fig. 2 where we also show the loss of models on the LQ set. As explained in the text, the loss on a domain is a good approximation of the KL divergence between the domain and the training domain, up to an additive constant (see paragraph “Loss on the HQ set as a proxy for the distance between CQF and HQ set.”). We observe that while the distance between CQF and HQ set is often a U curve with k, it is not the case for the distance between CQF and LQ set, which increases as we decrease k. In simple words, as the filtering gets more selective, the resulting filtered data is less LQ-aligned, which is expected.

---

### Official Review · Reviewer_kWia · 2025-11-02

**Soundness:** 2
**Presentation:** 2
**Contribution:** 2
**Rating:** 4
**Confidence:** 4

**Summary:**

This paper aims to explore the mechanism behind classifier-based quality filtering (CQF) that is commonly used for large-scale pretraining data. The authors question whether CQF selects high-quality (HQ set) data. The paper finds that although CQF improved the downstream metrics, it does not necessarily lead to better language modeling on the HQ set. Finally, the authors propose a framework to "evaluate" CQF using "data conditioning," where they find CQF captures properties more closely related to stylistic or domain similarities.

**Strengths:**

1. CQF is a commonly used method to curate a pre-training dataset for many state-of-the-art LLMs. Works aims to understand CQF is highly relevant and important.

2. The paper includes a large set of experiments ranging over multiple datasets and settings.

**Weaknesses:**

1. The paper challenges that CQF correlates with the data property of "universal quality"; however, it is unclear what "universal quality" means. There are no actionable findings pointed out from the paper, rather than showing that CQF works to better align with the downstream tasks. Is this not expected when HQ data is used from downstream targets?

I would be interested in creating a universally high-quality dataset -- without using directly downstream task datasets, and maybe using human/LLM annotations -- and exploring CQF using this data.

2. Evaluation tasks do not necessarily correlate with HQ data used in terms of the paper's analysis. Specifically, OpenOrca and OpenHermes datasets are instruction datasets, and OpenWebMath is a math domain data; however, there is no instruction-style or math evaluation. Supporting this, the downstream metric vs loss relationship is only different for ARC Eacy, where the evaluation and HQ set closely match.

3. Training details (token budget, model size, hparams) are not present in the main pages of the paper, which is important because pretraining dynamics are highly dependent on these factors. I would like to see a baseline where the model is trained with the base dataset and see if it performs above random in downstream evaluations.

**Questions:**

Please see `Weaknesses` for my questions.

---

> ### Author Response · Authors · 2025-11-24
>
> Dear reviewer,
>
> We thank you for your review and for taking time to read our paper. We are happy to hear that you found the topic “highly relevant and important” and that we conducted extensive experiments.
>
> We now answer the questions you have raised.
>
> >  it is unclear what "universal quality" means
>
> Indeed, as said in the paper, it is hopeless to define one true quality axis. We made this point explicit in Appendix D, where we can see no HQ set is superior to any other, all things considered. We provide one possible definition in sec.6, but we do not claim that it is the only way to define data quality.
>
> > There are no actionable findings pointed out from the paper, rather than showing that CQF works to better align with the downstream tasks.
>
> The purpose of the paper is to show that CQF aligns with the downstream task for reasons that were previously ignored by the community. We believe that providing a clear view of the inner workings of a method widely used for pretraining data filtering is a valuable contribution in itself. Moreover, we present one way to define quality, which, as pointed out by reviewer vK32, “represents a significant conceptual advance”
>
> Finally, as explained in the general response, we added one experiment where we show that, in some cases, CQF might lead to better models than training on the HQ set itself, even if we had enough tokens in the HQ set to train models. We believe that this leads to the prospects of using CQF as a way to improve HQ sets themselves instead of, for instance, synthetically augmenting them.
>
> >  Is this not expected when HQ data is used from downstream targets?
>
> This is expected, and this has been reported in several works. However, to the best of our knowledge, the reason why this happens was unknown to the community and providing this explanation is precisely what our paper does by challenging the fact that CQF selects data close to the HQ set.
>
> > I would be interested in creating a universally high-quality dataset \-- without using directly downstream task datasets, and maybe using human/LLM annotations \-- and exploring CQF using this data.
>
> This is an interesting point that could be further investigated for future work. However, we would like to clarify that one of the HQ sets we analyse in this paper already comes directly from human annotations (ELI5 from reddit), and some other from LLM annotations (OpenHermes from gpt4, OpenOrca from gp3.5 and gpt4). Please see table 1 for an exhaustive list of sources our paper covers. This paper does not focus on the difference in behavior of CQF from human data VS llm generated data but we agree with the reviewer that it would be very interesting to look into. Thanks for raising this point!
>
> > Evaluation tasks do not necessarily correlate with HQ data used in terms of the paper's analysis. Specifically, OpenOrca and OpenHermes datasets are instruction datasets, and OpenWebMath is a math domain data; however, there is no instruction-style or math evaluation.
>
> We used HQ sets that reflect the standard practice in the community, for instance, OpenHermes is used as HQ set in DCLM classifier, for DCLM [1] and Nemotron-CC [2] datasets, which are widely used pre-training datasets. Even though these datasets are not perfectly aligned with the downstream tasks as pointed out by the reviewer, using them for CQF improves downstream tasks, as reported many times in the literature (see related work section).
>
> [1] Li, J., Fang, A., Smyrnis, G., Ivgi, M., Jordan, M., Gadre, S.Y., Bansal, H., Guha, E., Keh, S.S., Arora, K. and Garg, S., 2024. Datacomp-lm: In search of the next generation of training sets for language models. Advances in Neural Information Processing Systems, 37, pp.14200-14282.
>
> [2] Su, D., Kong, K., Lin, Y., Jennings, J., Norick, B., Kliegl, M., Patwary, M., Shoeybi, M. and Catanzaro, B., 2025, July. Nemotron-cc: Transforming common crawl into a refined long-horizon pretraining dataset. In Proceedings of the 63rd Annual Meeting of the Association for Computational Linguistics (Volume 1: Long Papers) (pp. 2459-2475).
>
> > Training details (token budget, model size, hparams) are not present in the main pages of the paper.
>
> They were in the appendix; we have added them to the main text.
>
> > I would like to see a baseline where the model is trained with the base dataset and see if it performs above random in downstream evaluations.
>
> We made it clearer in the caption of Figure 2 that this baseline is already visible in the current figures, since it corresponds to CQF with k=100% (for instance, the leftmost point on the curves in fig.2). We indeed see better than random accuracy here (random would be 25% as this downstream task consists of multiple choices  with 4 choices).

---

### Author Response · Authors · 2025-11-24
**General Response**

Dear reviewers,

We thank you for your work and for having spent time rating our paper. We are happy to read that you found the topic “highly relevant and important”, that the experiments are extensive (R.kWia, dHu5 5AvM) and that we “challenge long-standing assumptions” on current SOTA data processing pipelines with “significant conceptual advances (R.vK32 and 5AvM). We summarize here the main practical contributions of our paper:

* **Task-aligned dataset filtering and selective filtering insights:** We show that CQF-selected data progressively align pretraining data with downstream benchmarks, while also revealing the HQ loss paradox where extreme filtering can underrepresent parts of the HQ distribution. We also clarify the distinction between importance sampling methods and CQF. These analyses provide practical guidance for curating datasets and balancing selectivity to maximize downstream performance.

* **CQF as a practical strategy beyond HQ augmentation:** We show that models trained on carefully filtered LQ data via nested CQF can outperform models trained on the HQ set itself, even when the HQ set has sufficient tokens. This demonstrates a practical approach for improving pretraining efficiency and quality without costly HQ data augmentation.

* **Introducing optimization-driven dataset quality:** We propose data conditioning, defining quality as a dataset’s ability to accelerate learning. Our scaling experiments show that this notion can be estimated using small-scale proxy models, giving practitioners a practical tool to evaluate dataset usefulness before large-scale training.

We have integrated the reviewers’ feedback directly into the main paper and address here the two main points that seem to be causing hesitation regarding its acceptance.

Some reviewers argued about

* the lack of actionable findings in the paper (R.kWia, vK32, 5AvM). While we believe that understanding the inner workings of a widely used method has merits, we have also added a novel experiment in section 5\. In that experiment, **we identify a case where training a model on the HQ set leads to worse performance than on the corresponding CQF sets**, *even though the HQ set has enough tokens to train without repetitions*. We believe that this experiment paves the way to using CQF to improve upon the HQ sets. By challenging the widespread assumption that simply adding more HQ data is always the best course of action, it offers actionable and practically significant insights that directly question current community efforts focused on HQ data augmentation.

* how impractical our definition of data-conditioning might be (R.kWia, vK32, 5AvM). We would like here to clarify the purpose of our last section where we (i) introduced a sound definition of data quality (ii) derived scaling laws. The scaling laws show consistent behavior with respect to model size and token counts. **This indicates that our notion of data quality can be effectively approximated with small-scale experiments as proxies.** We have made this point clearer in the text thanks to your feedback.

We hope this clarifies our paper and that the reviewers will reconsider their appreciation of our work in light of these explanations and the further experiments we provided.

---

### Author Response · Authors · 2025-12-03
**Summary and thank you**

Dear reviewers and AC,

We would like to thank you once again for the time and effort you dedicated to reviewing our paper. As the rebuttal period is over, we decided to summarize below the key contributions of our work to help streamline the AC’s decision process. In this paper,

- We provided a **principled analysis** of a current state-of-the-art data filtering pipeline, namely Classifier-Based Quality Filtering (CQF), which learns a notion of “quality’’ from a high-quality reference set. We highlighted an interesting unknown paradox in how data is filtered from the pretraining set, that is, in a way that improves performance on common benchmarks whilst at the same time deteriorating language modeling on the high quality set of reference. This makes CQF clearly contrast from importance sampling methods that try to mimic the distribution of the high quality set of reference. Yet, CQF works better than importance sampling in practice, hence the motivation for understanding better its inner workings.

- Our observations are supported by **comprehensive empirical validations across settings**. Indeed, we empirically ablated the phenomena at stake on a range of model sizes ranging from 125M to 1.3B, for multiple quality selection ratios, across different high quality sets of reference and training regimes (we performed both pertaining and finetuning experiments). The high quality sets of reference spans both standard pretraining corpora as well as instruction following, human generated and synthetic ones. Importantly, we control for data-scarcity effects by holding the total number of training tokens constant across all selection ratios and setups, focusing solely on the "quality" aspect.

- **We highlighted the practical relevance of our findings through additional rebuttal-phase experiments.** Based on the rebuttal, we have made clearer why our results are practical by providing further experiments based on some reviewer's questions. More specifically, we showed that training on the filtered set itself, CQF, can be better than training on the high quality set of reference, even if it came with infinite tokens. This offers a very practical approach for improving pretraining efficiency and quality without relying on costly data augmentation of the high quality set of reference. Given the substantial resources typically devoted to data augmentation, we believe this finding provides direct and highly practical insight.

- Finally **we proposed a definition for data quality** (at the dataset level) that places the optimization process at its core and provided extensive scaling laws to show that this notion **can be efficiently approximated in practice** by running small-scale proxies.

Although the reviewers did not have the opportunity to respond during the rebuttal window, we hope our edits, detailed responses, and additional experiments addressed their concerns adequately.

Best,

The authors

---

### Meta-Review · Area_Chair_K9Dw · 2026-01-07

**Summary:**

The paper's motivation is good, but the findings are not surprising, since it is well-known that "HQ" data has its limitations and it is usually helpful to include some other selected data in addition to what is designated as "HQ". The work is mostly diagnostic. Although the authors mention that such diagnosis could help further actions, it unclear what such actions could be besides that CQF and HQ could be combined (which is obvious since it is common to mix a number of datasets selected with different methods).

**Reviewer Concerns:**

The authors provided answers to all the reviewers' questions, but I believe the concern about actionability and probably some other concerns have not been fully addressed.

**Reviewer Scores:**

Some reviewers would maintain their negative opinion about this paper.

---

### Decision · Program_Chairs · 2026-01-26

Reject